# Evaluation of Adult Height in Patients with Non-Permanent Idiopathic GH Deficiency

Agnese Murianni [1], Anna Lussu [1], Chiara Guzzetti [1], Anastasia Ibba [1], Letizia Casula [1], Mariacarolina Salerno [2], Marco Cappa [3] and Sandro Loche [1,*]

1   SSD Endocrinologia Pediatrica e Centro Screening Neonatale, Ospedale Pediatrico Microcitemico "A. Cao", 09121 Cagliari, Italy
2   U.O.S. di Endocrinologia Pediatrica, Azienda Ospedaliera Universitaria "Federico II", 80131 Napoli, Italy
3   U.O.C. di Endocrinologia, IRCCS Ospedale Pediatrico Bambino Gesù, 00163 Roma, Italy
*   Correspondence: sandro.loche@aob.it; Tel.: +39-070-5296-7904

**Abstract: Background:** Several studies have evaluated the role of IGF-1 in the diagnosis of growth hormone deficiency (GHD). According to a recent study, an IGF-1 concentration of a $-1.5$ standard deviation score (SDS) appeared to be the best cut-off for distinguishing between children with GHD and normal children. This value should always be interpreted in conjunction with other clinical and biochemical parameters for the diagnosis of GHD, since both stimulation tests and IGF-1 assays have poor diagnostic accuracy by themselves. Our study was designed to evaluate the adult height (AH) in children with short stature and baseline IGF-1 concentration $\leq -1.5$ SDS. **Design:** This retrospective analysis included 52 children and adolescents evaluated over the last 30 years for short stature and/or deceleration of the growth rate who underwent diagnostic procedures to evaluate a possible GHD. Only the patients who had baseline IGF-1 values $\leq -1.5$ SDS at the time of the first test were included in the study. Patients with genetic/organic GHD or underlying diseases were not included. **Method:** The case group consisted of 24 patients (13 boys and 11 girls) with non-permanent, idiopathic, and isolated GHD (peak GH < 10 µg/L after two provocative tests with arginine (Arg), insulin tolerance test (ITT), and clonidine (Clo), or <20 µg/L after GHRH + Arginine (GHRH+Arg); normal MRI; normal GH; and/or normal IGF-1 concentrations at near-AH). These patients were treated with GH (25–35 µg/kg/die) until near-AH. The control group consisted of 28 patients (23 boys and 5 girls) with idiopathic short stature (ISS, normal peak GH after provocative testing, no evidence of other causes for their shortness). Both groups had basal IGF-1 $\leq -1.5$ SDS. **Results:** AH and height gain in both groups were comparable. In the group of cases, mean IGF-1 SDS at the time of diagnosis was significantly lower than the levels found at the time of retesting. **Conclusions:** In this study, both treated patients with idiopathic GHD and untreated patients with ISS reached similar near-AHs (within target height) and showed similar increases in SDS for their height. Thus, the efficacy of treatment with rhGH in these patients may be questionable. This could be due to the fact that children with ISS are frequently misdiagnosed with GHD.

**Keywords:** short stature 1; growth hormone deficiency 2; insulin-like growth factor-1 3

## 1. Introduction

Growth hormone deficiency (GHD) is caused by impaired production and/or secretion of growth hormone (GH), with resulting growth failure. The incidence of GHD ranges from 1:4000 to 10,000 and is frequently isolated. GHD can be idiopathic, congenital, or acquired (by trauma, tumors, infections, radiation therapy, etc.). Mutations in the genes encoding for GH or its receptor can cause GHD or GH resistance. The clinical phenotype is more complex when genes encoding from transcription factors are involved (such as SOX3, HESX1, GLI2, OTX2, LHX3, LHX4, PROP1, and POU1F1). These mutations are usually associated with multiple pituitary hormone deficiencies and other extra pituitary abnormalities [1].

The diagnosis of GHD is based on auxological, biochemical, neuro-radiological, and genetic investigations. GH secretion is pulsatile and regulated by multiple physiologic factors, including age, puberty, nutritional status, and body weight. Its secretion is commonly evaluated with the use of stimulation tests. In the past 50 years, a number of different stimulation tests for GH secretion have been suggested. The cut-off threshold was initially set at 5–7 g/L, and was arbitrarily raised to 7–10 thereafter. The selection of this cut-off level, however, did not consider the type of stimulus nor the variability of responses due to gender, age, puberty, body mass index (BMI), or other factors such as non-physiological test techniques and type of assay, thus making it difficult to interpret the results. It has become clear that provocative tests have poor reproducibility, specificity, and sensitivity [2].

Spontaneous and stimulated GH secretion are markedly affected by sex steroids. GH secretion increases during spontaneous pubertal maturation or after the exogenous administration of sex steroids, and this phenomenon is principally due to the action of estrogens. False positive responses to GHST are frequently observed at the peripubertal age, when sex steroid concentrations are physiologically low [3]. In this period of life, it is difficult to discriminate GHD from constitutional delay of growth and puberty (CDGP). The administration of sex steroids before GHST (priming) could be useful, according to some authors [3,4], to differentiate GHD from CDGP. However, it is still debated whether prepubertal children should be "primed" with sex hormones prior to GHST. Supraphysiologic levels of sex hormones may artificially increase the release of endogenous growth hormone. This overestimation could lead to erroneous negative results and prevent eligible children from receiving growth hormone therapy [5].

The most reliable marker of GH action is insulin-like growth factor-1 (IGF-1) [6]. Several studies have evaluated the role of IGF-1 in the diagnosis of GHD over the past decades [7,8], but the results are often not comparable [7]. Overall, the results indicate that IGF-1 has good specificity (about 90%) and low sensitivity (about 70%), indicating that subnormal levels of IGF-1 ($<-2$ SDS) are highly predictive of GHD, but that normal levels do not always rule out GHD. In the context of GHD, the IGF-1 cut-off level is still a matter of discussion between scientific societies, and the standard deviation score (SDS) currently ranges between $-1.5$ and $-2$ [7,9,10]. According to a recent study, the best cut-off for discriminating patients with GHD from healthy subjects is $-1.5$ SDS [11]. However, this value should always be interpreted together with other clinical and biochemical parameters. Some authors [5,12–14] have compared auxological factors, such as the predicted final height, in GHD and idiopathic short stature (ISS) patients. ISS is a condition in which the height of an individual is more than 2 SD scores (SDS) below the corresponding mean height for a given age, sex, and population group without evidence of systemic, endocrine, nutritional, or chromosomal abnormalities. Particularly, subjects with ISS have normal birth weights and normal GH responses to stimulation testing. Some studies have found that near-adult heights (AH) were close to target heights (TH) in both ISS and GHD patients. The fact that near-AH in both groups was close to TH indicates that the patients reached their genetic potential [5].

The aim of our study was to evaluate the AH in short patients with and without GHD with baseline IGF-1 concentrations of $\leq-1.5$ SDS.

## 2. Subjects and Methods

### 2.1. Subjects

This retrospective analysis included 52 children and adolescents (36 males and 16 females aged 3.8–15.9 years) who were evaluated for short stature and/or deceleration of the growth rate in 3 Pediatric Endocrinology Units in Italy (Cagliari, Naples, and Rome) over the last 30 years. All subjects included in the study underwent diagnostic procedures to confirm GHD. The following information was gathered throughout the clinical visits, both at the time of diagnosis and during any subsequent re-evaluation: height (HT), weight, pubertal stage, BMI, height velocity (HV), target height (TH), and, when possible, AH. The whole cohort was, therefore, divided into 2 groups of subjects: GHD (cases) and

non-GHD (controls). The case group consisted of 24 patients (13 boys and 11 girls) with non-permanent, idiopathic, and isolated GHD (low GH peak after two provocative tests, normal MRI, normal GH and/or IGF-1 concentrations at near-AH). The control group consisted of 28 patients (23 boys) with ISS, normal GH peak after the provocative test, and no evidence of other causes for their shortness. Children with other underlying diseases, as well as those with genetic or organic GHD, were excluded from the study.

### 2.2. Methods

BMI was calculated with the weight/height squared formula [15]. Pubertal stage was assessed according to the Tanner classification [16]. The HV was calculated with the following formula: [(height time 2-height time 1)/12 × (elapsed months between visit 1 and visit 2)]. Growth rate was measured at intervals >6 months. The growth rate was calculated before performing the stimulation tests (pre-test HV) and, respectively, one year after starting therapy with rhGH in the patients with IGHD and one year after the tests in the controls (HV post-test). Near-AH (growth rate <1–2 cm/year) was calculated during follow-up, and patients who missed appointments were contacted again years later, including during the COVID-19 pandemic, to complete the final evaluations. Because of the COVID-19 emergency, measurements of AH in 27 controls were taken either at home (following the instructions received by the centers, i.e., in an upright position, barefoot, feet together, with the heels and the head adhering to the wall by passing a virtual line parallel to the floor between the lower edge of the orbit and the external auditory canal) or in the pharmacy, and then communicated to our Center. For auxological data, SDS was calculated based on Italian references [17]. Bone age (BA) was assessed using the TW20 method or the comparative method of Greulich and Pyle [18]. Basal IGF-1 and insulin growth factor binding protein-3 (IGFBP-3) were also measured. The SDS was calculated using the references for the method.

Assessment of GH secretion was performed using the clonidine (Clo), arginine (Arg), insulin tolerance (ITT), and GHRH+Arg stimulation tests. Arginine was given intravenously (0,5 g/kg, max 30 g) for 30 min, and GH levels were measured at 30, 0, 15, 30, 45, 60, 90, and 120 min. Insulin was administered intravenously (0.05–0.1 U/kg) and GH and glucose levels were determined at 0, 30, 60, 90, and 120 min. Clonidine was administered orally (0.15 mg/m2), and blood samples for GH determination were collected at 0, 30, 60, 90, and 120 min. GHRH was given intravenously (1 μg/kg) followed by coadministration of arginine 0.5 g/kg; blood samples were taken every 15 min until +90 min. A GH peak $\geq 10$ μg/L was considered as the limit for all tests, except for the enhanced GHRH+Arg test, whose normal limit is defined by a GH peak of 20 μg/L. The GH peak, in response to different stimuli, was analyzed individually for each test, as well as overall for the Clo, Arg, and ITT tests (peak GH). Sex hormone priming was never used [2]. All patients who performed poorly on the first stimulation test underwent a second stimulation test (at least 2 days afterward). All tests were carried out in the morning after fasting overnight (8–9 am) [19]. In all patients, a diagnosis of GHD was established before 2014 in accordance with the international consensus [12]. Only patients who had baseline IGF-1 values $\leq -1.5$ SDS at the time of the first test were included in the retrospective study. Patients diagnosed with IGHD received rhGH replacement therapy at a dose of 25–35 μg/kg/day [20]. Five patients were retested (GHRH+Arg), and the IGF-1 value was monitored in nine patients during the transition stage. In none of the patients was it necessary to continue rhGH therapy after achieving AH.

### 2.3. Assays

Serum levels of GH and IGF-1 were determined by immunofluorescence (Immulite 2000; Diagnostic Products Corp., Los Angeles, CA). The sensitivity of the assay was 0.01 μg/L for GH and 2.6 nmol/L for IGF-1. The intra- and inter-assay coefficients of variation for GH were, respectively, 4.2–6.6 and 2.9–4.6% for GH values between 2.6 and

17 µg/L; for the IGF-1, the intra- and inter-assay coefficients of variation were 3.4% and 7.1%, respectively.

*2.4. Statistical Analysis*

The Kolmogorov–Smirnov test was used to evaluate the distribution of the data. All data were normally distributed except for age, IGF-1 SDS, GHRH+Arg peak, GH peak, and BA. Comparisons between the groups were performed using Student's t-test and Mann–Whitney's *U*-test for normally distributed and non-normally distributed variables, respectively. All values were reported as medians and ranges (continuous variables) or as counts and percentages (categorical variables). A *p*-value < 0.05 (two-tailed) was considered significant. All statistical calculations were performed using Graph Pad Prism software, version 6 (GraphPad Software Inc., La Jolla, CA, USA).

**3. Results**

The IGHD group (cases) included 24 patients: 13 males (54.2%) and 11 females (45.8%). The non-GHD group (controls) included 28 patients: 23 males (82.2%) and 5 females (17.8%). The baseline HT SDS, age, gender, pubertal status, pre-test HV SDS, IGF-1 SDS, IGFBP-3 SDS, and BA were similar in the 2 groups (Table 1, Figure 1). The higher BMI SDS values found in the cases compared to the controls were statistically significant (*p* = 0.0003). Only two of the patients in the cases group had started puberty before rhGH treatment; the others began puberty during rhGH treatment. The post-test HV SDS value and the TH SDS were significantly higher in the cases than in the controls (*p* = 0,04 and *p* = 0.017, respectively). The average duration of therapy was 4.5 years (3.5; 9.5), lasting until AH was reached. AH and height gain in both groups were comparable (Figure 2). AH was measured at 17.3 (14.8; 20) years old in cases and at 16.9 (14.1; 28.2) years old in controls. The median IGF-1 SDS at diagnosis in the cases was −2.13 (−5.48; −1.56), which was significantly lower than that observed at re-evaluation upon reaching AH (−0,01 (−1.99; 2.66); *p* = 0.002).

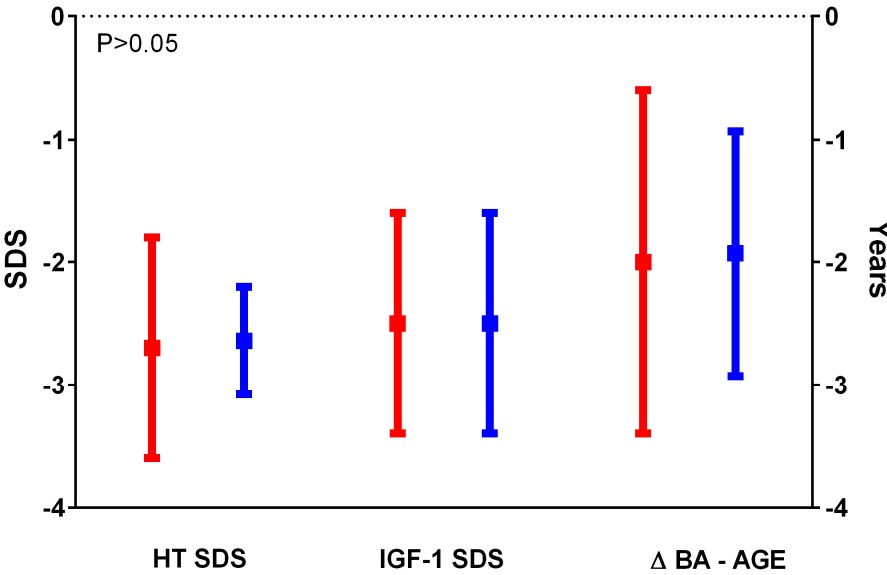

**Figure 1.** Comparison between HT SDS, IGF-1 SDS, and ΔBA-AGE SDS in the cases (**in red**) and in the controls (**in blue**); data were comparable between the 2 groups.

**Table 1.** Clinical characteristics of patients, divided into cases and controls. All values are reported as medians (ranges) or numbers (percentages).

|  | CASES (n = 24) | CONTROLS (n = 28) | *p* |
|---|---|---|---|
| SEX M/F | 13 (54.2%)/11 (45.8%) | 23 (82.2%)/5 (17.8%) | 0.1 |
| AGE (years) | 12.08 (3.75; 16.16) | 12.46 (5.00; 15.90) | 0.61 |
| HT SDS | −2.65 (−4.50; −0.01) | −2.60 (−3.58; −1.78) | 0.58 |
| TH SDS | −1.00 (−2.55; 0.30) | −1.60 (−2.45; 0.00) | **0.017** |
| BMI SDS | −0.02 (−2.87; 2.05) | −1.31 (−2.91; 0.38) | **0.0003** |
| Prepubertal/pubertal | 19 (79.2%)/5 (20.8%) | 15 (53.5%)/13 (46.4%) | 0.08 |
| Puberty onset (years) | 12.8 (9.11; 15.3) | 12.9 (11.9; 14.11) | 0.87 |
| BA (years) | 10.40 (3.90; 13.90) | 10.35 (3.00; 15.00) | 0.54 |
| Δ BA—Age (years) | −2.20 (−4.20; −0.10) | −1.87 (−3.70; 0.30) | 0.91 |
| HV SDS pre-test | −1.11 (−3.79; 1.11) | −2.77 (−6.91; −0.46) | 0.05 |
| HV SDS post-test | 2.47 (−3; 11.64) | −1.01 (−5.7; 6.9) | **0.04** |
| IGF-1 SDS | −2.13 (−5.48; −1.56) | −2.19 (−4.9; −1.55) | 0.87 |
| IGFBP-3 SDS | −0.65 (−3.08; 0.37) | −0.61 (−1.87; 0.77) | 0.43 |
| Peak GH Arg (μg/L) | 6.78 (3.00; 7.90)<br>(n = 11) | 13.25 (6.42; 21.20)<br>(n = 4) | **0.0021** |
| Peak GH ITT (μg/L) | 5.70 (3.60; 7.91)<br>(n = 5) | 8.56 (8.56; 8.56)<br>(n = 1) | n.a. |
| Peak GH Clo (μg/L) | 5.48 (0.07; 9.80)<br>(n = 22) | 13.20 (4.42; 28.40)<br>(n = 16) | **<0.0001** |
| Peak GH (μg/L) | 7.00 (3.00; 9.80)<br>(n = 24) | 13.95 (8.56; 28.40)<br>(n = 20) | **<0.0001** |
| Peak GHRH+Arg (μg/L) | 6.89 (6.18; 8.60)<br>(n = 3) | 26.60 (14.60; 40.00)<br>(n = 10) | **0.007** |
| AH SDS | −1.56 (−3.80; 1.76) | −1.76 (−3.44; 1.00) | 0.3 |
| Δ AH SDS- HT SDS | 1.29 (0.07; 3.78) | 0.95 (−1.27; 3.20) | 0.08 |
| Δ AH SDS- TH SDS | −0.31 (−3.40; 2.76) | −0.16 (−1.97; 2.00) | 0.69 |

Arg, arginine; BMI, body mass index; Clo, clonidine; AH, adult height; GH, growth hormone; GHRH, growth hormone-releasing hormone; HT, height; HV, height velocity; IGF-1, insulin-like growth factor 1; IGFBP-3, insulin growth factor binding protein 3; ITT, insulin tolerance test; n.a., not available; SDS, standard deviation score; TH, target height; GH peak = GH peak after Arg, ITT, or Clo. We used bold to underline significant results.

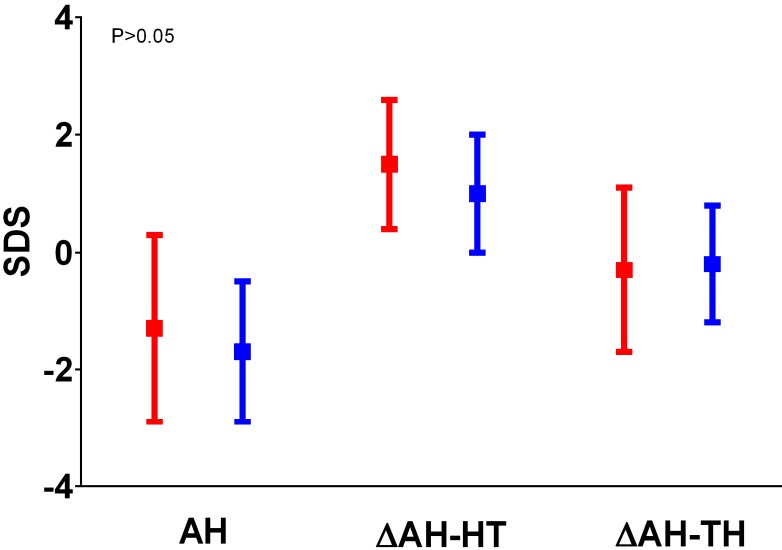

**Figure 2.** Comparison between AH, ΔHT SDS, and ΔAH SDS-TH SDS in the cases (**in red**) and in the controls (**in blue**); data were comparable between the 2 groups.

Finally, the 16 patients with partial GHD (peak GH 5–10 µg/L) and the 8 patients with severe GHD (peak < GH 5 µg/L) were analyzed separately (Table 2). The results were similar to those obtained comparing the 24 cases and controls (higher BMI SDS value, TH SDS (partial GHD), HV SDS post-test, and lower peak GH on the Clo test and Arg+GHRH test in cases).

**Table 2.** Clinical characteristics of patients, divided into Partial GHD (peak GH 5–10 µg/L), severe (peak < GH 5 µg/L), and controls. All values are reported as medians (ranges) or numbers (percentages). We used bold to underline significant results.

| | Controls (n = 28) | Partial GHD (n = 16) | *p* (Partial GHD vs. Controls) | Severe GHD (n = 8) | *p* (Severe GHD vs. Controls) |
|---|---|---|---|---|---|
| SEX M/F | 23 (82.2%)/5 (17.8%) | 9 (56.25%)/7 (43.75%) | 0.05 | 4 (50%)/4 (50%) | 0.08 |
| AGE (years) | 12.46 (5.00; 15.90) | 11.50 (3.75; 15.66) | 0.31 | 12.63 (7.08; 16.16) | 0.59 |
| HT SDS | −2.60 (−3.58; −1.78) | −2.80 (−4.50; −2.00) | 0.12 | −2.45 (−3.87; −0.01) | 0.38 |
| TH SDS | −1.60 (−2.45; 0.00) | −1.00 (−2.41; 0.20) | **0.03** | −0.89 (−2.55; 0.3) | 0.07 |
| BMI SDS | −1.31 (−2.91; 0.38) | −0.11 (−2.87; 1.41) | **0.04** | 0.06 (−1.35; 2.05) | **0.0006** |
| Prepubertal/ pubertal | 15 (53.5%)/13 (46.4%) | 12 (75%)/4 (25%) | 0.2 | 7 (87.5%)/1 (12.5%) | 0.11 |
| Puberty onset (years) | 12.9 (11.9; 14.11) | 12.40 (9.11; 15.30) | 0.64 | 13.85 (12.8; 14.9) | 0.19 |
| BA (years) | 10.35 (3.00; 15.00) | 10.40 (3.90; 13.90) | 0.57 | 10.5 (9.7; 11.3) | 0.79 |
| Δ BA—Age (years) | −1.87 (−3.70; 0.30) | −1.40 (−4.20; −0.10) | 0.99 | −2.25 (−2.3; −2.2) | 0.65 |
| HV SDS pre-test | −2.77 (−6.91; −0.46) | −1.03 (−3.79; −0.29) | 0.17 | −1.2 (−1.84; 1.11) | 0.07 |
| HV SDS post-test | −1.01 (−5.7; 6.9) | 2.47 (−3.00; 10.23) | **0.01** | 4.78 (0.17; 11.64) | **0.003** |
| IGF-1 SDS | −2.19 (−4.9; −1.55) | −2.08 (−5.48; −1.56) | 0.94 | −2.61 (−3.61; −1.70) | 0.52 |
| IGFBP-3 SDS | −0.61 (−1.87; 0.77) | −0.57 (−3.08; 0.37) | 0.42 | −1.89 (−2.69; −1.1) | 0.12 |
| Peak GH Arg (µg/L) | 13.25 (6.42; 21.20) (n = 4) | 6.84 (4.80; 7.90) (n = 10) | 0.054 | 3 (n = 3) | n.a. |
| Peak GH ITT (µg/L) | 8.56 (8.56; 8.56) (n = 1) | 5.75 (4.27; 7.91) (n = 4) | n.a. | 3.6 (n = 1) | n.a. |
| Peak GH Clo (µg/L) | 13.20 (4.42; 28.40) (n = 16) | 6.60 (3.70; 9.80) (n = 14) | **<0.0001** | 2.25 (0.07; 3.93) (n = 8) | **<0.0001** |
| Peak GH (µg/L) | 13.95 (8.56; 28.40) (n = 20) | 7.15 (5.80; 9.80) (n = 14) | **<0.0001** | 2.70 (−0.07; 3.93) | **<0.0001** |
| Peak GHRH+Arg (µg/L) | 26.60 (14.60; 40.00) (n = 10) | 6.53 (6.18; 6.89) (n = 2) | **0.03** | 8.6 (n = 1) | n.a. |
| AH SDS | −1.76 (−3.44; 1.00) | −1.70 (−3.34; 1.76) | 0.78 | −0.28 (−3.8; 1.1) | 0.07 |
| Δ AH SDS− HT SDS | 0.95 (−1.27; 3.20) | 1.16 (0.10; 3.78) | 0.49 | 1.85 (0.07; 2.73) | 0.07 |
| Δ AH SDS− TH SDS | −0.16 (−1.97; 2.00) | −0.77 (−3.40; 2.76) | 0.18 | 0.39 (−1.8; 1.99) | 0.44 |

## 4. Discussion

In our retrospective study, we evaluated the AH of 24 IGHD patients treated with GH and 28 untreated controls with ISS. In both groups, the baseline IGF-1 concentrations were ≤−1.5 SDS. Despite low and similar IGF-1 levels at the time of diagnosis, both groups reached their genetic target heights. Therefore, it is conceivable that ISS patients would not have benefited from rhGH treatment. Another possibility is that some of the patients were incorrectly diagnosed as GHD, and probably would have also reached their TH if left untreated. HT SDS, TH SDS, age, gender, pubertal status, pre-test HV SDS, IGF-1 SDS, IGFBP-3 SDS, and BA were similar in the two groups. The BMI SDS in the cases was higher than in the controls (*p* = 0.0003), confirming the already-known inverse relationship between BMI and GH response to standard stimulation tests [21]. As expected, based on previous studies [22], the post-test HV SDS values were significantly higher in the group of

cases treated with rhGH than in the controls ($p = 0.04$). AH and height gain in both groups were comparable, as previously described [22–24]. Puberty onset may have contributed to the rise in post-test HV SDS that we observed in both cases and controls, as the median age of pubertal onset was recorded a few months after the GHST was performed.

It was shown that 85% of patients with IGHD had normal GH secretion when the test was repeated 1–6 months after treatment was stopped, confirming the difficulty of differentiating between IGHD and ISS [5]. Bang et al. [25] further confirmed the difficulties of differentiating between the two groups at the time of diagnosis. Based on the afore-mentioned data, the only value that was useful in distinguishing the group of children with GHD from those with ISS was the GH cut-off after stimulus test, arbitrarily accepted below <10 µg/mL. However, the established value of the above-mentioned cut-off differed across different countries, varying from 6 to 10 µg/mL, depending, among other factors, on the immune assay used [26,27]. Some studies that have considered auxological parameters, including AH, indicate that there are many more similarities between children with IGHD and ISS than those with severe GHD [5,12]. Therefore, we can assume that the current diagnostic approaches for the diagnosis of GHD could provide imprecise results, as demonstrated in a recent study by Ariza-Jimenez et al. [13] Graber et al. [28] suggest in their article that the patient's entire clinical profile, including not only the results of the stimulation test, but also their anthropometric measurements, should be taken into account, including HV, physical findings, screening tests, and IGF-1 and IGF-BP3. Furthermore, Ibba et al. [20] showed that an IGF-1 value of −1.5 SDS was the best cut-off to discriminate GHD patients from controls. This study has shown that IGF1 measurement has poor sensitivity, specificity, and efficiency (67, 62, and 67%, respectively) at a cut-off of 1.5 SDS, as evidenced by an AUC of 0.68. Higher specificity (77.2%), but significantly lower sensitivity (46.8%), was observed at an IGF-1 cut-off of 2 SDS, as recommended by the current guidelines. The findings of this study further demonstrate that the results of IGF1 measurement for the diagnosis of GHD must be interpreted together with other clinical and biochemical findings. In fact, they indicate that the combined evaluation of IGF1 concentrations and peak GH after IGF1 improves test specificity, reaching 95.8% for ITT, 98.4% for the clonidine test, and 95.6% for the arginine test, and lowers the possibility of false positive results. IGF-1 is the main GH-dependent factor responsible for bone growth, although a number of other factors besides GH may influence IGF-1 serum levels, thus making the interpretation of the results difficult [29].

Growth hormone levels typically rise during puberty as a result of a greater volume of GH released during each pituitary secretory episode [30]. The concentration of sex hormones is one of the factors that affects the spontaneous or induced release of GH. Sexual steroids and GH concentrations are low during the peri-pubertal era. Patients' growth rates fall during this time, reaching levels that are even lower than those seen during prepuberty (4 cm/year). The rapid growth seen during puberty (8–10 cm/year) is caused by the subsequent and gradual rise in sex steroid levels, which increases spontaneous GH release. For this reason, normalization of GH secretion at the time of puberty occurs frequently in patients with isolated GHD, and retesting at the end of growth is necessary to confirm the diagnosis [31].

The exogenous administration of sex steroids also leads to an increase in the secretion of GH, mainly due to the action of estrogens [32]. According to the 2019 GH Research Society recommendations [33], peripubertal settings may benefit from sex steroid priming before GHST, in order to differentiate GHD from CDGP. Its application is still debatable; regarding the choice of patient, time, dose, preparation, and administration of sex steroids, there is no universal agreement. Priming could lead to an increase in the secretion of artificial and temporary GH, with a subsequent return to secretion of subnormal GH [34]. Endogenous growth hormone secretion may be exaggerated at these supraphysiologic doses of sex hormones. This overestimation may lead to false negative results and prevent growth hormone therapy from being given to eligible children [32]. Furthermore, the Clo test was used the most frequently in our investigation. Previous research [35,36] has shown

that there are no differences between prepubertal and pubertal patients using the Clo test, because it is unaffected by puberty. The low prevalence of false positive subnormal responses, as shown in the study by Ibba et al. [35], demonstrates that the Clo test is reliable in both prepubertal and pubertal children, reducing the need for sex steroid priming.

Our results are similar to those of other studies, showing that children with IGHD and ISS have comparable growth responses and do not seem to benefit from rhGH therapy [13,14,37]. Our findings are consistent with a previous study in which the sole distinguishing factor between the cases and controls was the GH peak, which was lower in GHD, despite the fact that the other parameters, such as AH, bone age, bone age delay, and height increase, were comparable [5]. Again, this could be related to the poor accuracy of stimulation tests and IGF-1 measurements. In contrast, a study by Kristom et al. [38] found that rhGH therapy improves the prognosis of AH and height gain in a dose-dependent manner in pre-pubertal children with ISS. Those patients were randomized into three groups: a group treated with standard doses of rhGH (33 µg/kg/day), a group treated with high doses of rhGH (67 µg/kg/day), and an untreated control group. The results of this study show that, in the two treated groups, the AH, the gain in stature and the increase in IGF-1 were greater than in the controls. However, these results are not reproducible in clinical practice. In Italy, according to note 39 of the AIFA (Italian Medicines Agency), treatment with rhGH is not authorized in patients with ISS and, in any case, the maximum dosage of somatotropin corresponds to 50 µg/kg/day, according to EMA (European Medicines Agency) recommendations.

This study confirms that some patients diagnosed as IGHD have ISS. In our series, the IGF-1 value at AH was normal in all patients diagnosed with IGHD (both in the five patients who underwent retesting and in the nine patients who were not retested). Additionally, in some cases, patients may have pubertal delay, further challenging a diagnosis of GHD [22]. Our study presents some limitations, including the small number of patients and its retrospective nature. Therefore, it was not possible to evaluate the IGF-1 value of the controls at follow-up. Furthermore, as discussed, measurements of AH in the control group were not taken in hospital, in some cases due to COVID restrictions.

## 5. Conclusions

In our study, both treated patients with idiopathic GHD (16 with partial and 8 with severe GHD) and untreated patients with ISS reached the same near-AH (within target height) and had similar gains in height and SDS. Thus, the efficacy of treatment with rhGH in these patients may be questionable.

**Author Contributions:** Conceptualization, methodology and validation: A.M., A.L., C.G., A.I., L.C., M.S., M.C. and S.L.; software: C.G.; formal analysis and investigation: A.M., A.L. and C.G.; data curation: A.M.; writing—original draft preparation and writing—review and editing: A.M., A.L., C.G. and S.L.; supervision: S.L. All authors have read and agreed to the published version of the manuscript.

**Funding:** This research received no external funding.

**Institutional Review Board Statement:** The study was conducted in accordance with Declaration of Helsinki, and approved by the Ethics Committee of Azienda Ospedaliera Universitaria, Cagliari (protocol cod PG/2019/8866 26/06/2019).

**Informed Consent Statement:** Informed consent was obtained from all subjects involved in the study.

**Data Availability Statement:** The data presented in this study are available on request from the corresponding author. The data are not publicly available due to privacy.

**Conflicts of Interest:** The authors declare no conflict of interest.

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
