# Peer review of "Evaluation of Adult Height in Patients with Non-Permanent Idiopathic GH Deficiency"

_endocrines, doi:10.3390/endocrines4010015_

Round 1

Reviewer 1 Report

Congratulations! This is a very interisting study. I have a question: If there was a group evaluated over the last 30 years, why the adult height was measured during the COVID-19 pandemic? Please, justify.

Some questions:

a) Is the pubertal stage not  evaluated in all patients? (Group 14, control 27 = table 1). Or is there an error? Please, clarify.

b)Was the puberty onset during the GH treatment in GHD group? Please, clarify.

c) I noted that HV SDS post-test is better than HV SDS pre-test in control patients. Maybe, the puberty started at that year, because they werw 12.46yr-old at baseline. Is it possible? Describe.

d) Cite the age at adult height, and the BMI SDS at that time.

e) Discussion: Please, discuss something about puberty, priming and the results on stimulation tests. As some patients were prepubertal at 12.08yr-old and the puberty onset was at 12.8, I think that the non-responsive GH on standard stimulation tests should be due to that.

Suggestions: Yau M, Rapaport R. Growth Hormone Stimulation Testing: To Test or Not to Test? That Is One of the Questions. Front Endocrinol (Lausanne). 2022 Jun 9;13:902364. doi: 10.3389/fendo.2022.902364. PMID: 35757429; PMCID: PMC9218712. 

Partenope C, Galazzi E, Albanese A, Bellone S, Rabbone I, Persani L. Sex steroid priming in short stature children unresponsive to GH stimulation tests: Why, who, when and how. Front Endocrinol (Lausanne). 2022 Nov 29;13:1072271. doi: 10.3389/fendo.2022.1072271. PMID: 36523598; PMCID: PMC9744763.

Minor details:

line 58 - final heights (FH)

line 59- ISS and GHD

Line 77- Please, complete the ISS definition. Ref. Cohen P, Rogol AD, Deal CL, Saenger P, Reiter EO, Ross JL, Chernausek SD, Savage MO, Wit JM.2007 ISS Consensus Workshop participants. Consensus statement on the diagnosis and treatment of children with idiopathic short stature: a summary of the Growth Hormone Research Society, the Lawson Wilkins Pediatric Endocrine Society, and the European Society for Paediatric Endocrinology WorkshopJournal of Clinical Endocrinology and Metabolism 2008. 93 4210–4217. ( 10.1210/jc.2008-0509)

Line 115- "Average duration of therapy was 4.5 years (3.5; 9.5)." It is a result, not a method. 

Line 200 - GH secretion when the test was repeated after 1-6 months treatment stop.

Line 214 - IGF-BP3. Correct, please.

Author Response

Responses to comments and queries of Reviewer 1:

Congratulations! This is a very interesting study. I have a question: If there was a group evaluated over the last 30 years, why the adult height was measured during the COVID-19 pandemic? Please, justify.

A comment has been added in lines 116-123  

Some questions:

  1. a) Is the pubertal stage not evaluated in all patients? (Group 14, control 27 = table 1). Or is there an error? Please, clarify.

We agree with the reviewer. We have modified the table 1 and the results accordingly.  

  1. b) Was the puberty onset during the GH treatment in GHD group? Please, clarify.

Only two of the patients in the cases group had started puberty before rhGH treatment; the others began puberty during the rhGH treatment. (Lines 170-171)

  1. c) I noted that HV SDS post-test is better than HV SDS pre-test in control patients. Maybe, the puberty started at that year, because they were 12.46yr-old at baseline. Is it possible? Describe.

We agree with the reviewer. We know that the median age at which puberty began for the 28 controls was 12.9 years while the median age at GH stimulation test was 12.5, therefore it's possible that an increase in post-test HV SDS in this group of patients would have been due to the onset of puberty. However, an effect of puberty on HV post-test could be hypothesized in cases (median age of test 12,08 years, and median age of pubertal onset 12,8 years). (Lines 236-238)

  1. d) Cite the age at adult height, and the BMI SDS at that time.

AH was measured at 17,3 (14,8;20) years old in cases and 16,9 (14,1;28,2) years old in controls. We have modified the results accordingly (Lines 173-177)

  1. e) Discussion: Please, discuss something about puberty, priming and the results on stimulation tests. As some patients were prepubertal at 12.08yr-old and the puberty onset was at 12.8, I think that the non-responsive GH on standard stimulation tests should be due to that.

Suggestions: Yau M, Rapaport R. Growth Hormone Stimulation Testing: To Test or Not to Test? That Is One of the Questions. Front Endocrinol (Lausanne). 2022 Jun 9;13:902364. doi: 10.3389/fendo.2022.902364. PMID: 35757429; PMCID: PMC9218712. 

Partenope C, Galazzi E, Albanese A, Bellone S, Rabbone I, Persani L. Sex steroid priming in short stature children unresponsive to GH stimulation tests: Why, who, when and how. Front Endocrinol (Lausanne). 2022 Nov 29;13:1072271. doi: 10.3389/fendo.2022.1072271. PMID: 36523598; PMCID: PMC9744763.

We agree that pubertal stage could affect test results, even though we are aware that these results are often imprecise and impacted by a number of other of factors. GH stimulation tests either with or without priming have significant drawbacks. Priming with sex steroids is not a conventional procedure, and, in fact is not widely used. Its routine application, although suggested in same cases, is still debated. There is no uniform agreement on the patients to be primed, time, dose, preparation, and administration of sex steroids. Priming could lead to a transient and artificial increase of GH, with subsequent return to subnormal GH release. Endogenous GH secretion may be exaggerated following these supraphysiologic doses of sex hormones. This overestimation may lead to false negative responses and prevent giving growth hormone therapy to eligible children. Clonidine testing, which has been shown in earlier investigations to be unaffected by puberty (Reliability of clonidine testing for the diagnosis of growth hormone deficiency in children and adolescents. Ibba A, Guzzetti C, Casula L, Salerno M, Di Iorgi N, Allegri AME, Cappa M, Maghnie M, Loche S.Clin Endocrinol (Oxf). 2018 Dec;89(6):765-770. doi: 10.1111/cen.13845. Epub 2018 Sep 25.PMID: 30171702), is the primary method employed in this study. The low proportion of false positive subnormal responses found in this study demonstrates that CT is reliable and effective in both pre-pubertal and pubertal children, and that sex steroid priming may not be necessary. (Lines 278-292).

line 58 - final heights (FH) Done

line 59- ISS and GHD Done

Line 77- Please, complete the ISS definition. Ref. Cohen P, Rogol AD, Deal CL, Saenger P, Reiter EO, Ross JL, Chernausek SD, Savage MO, Wit JM.2007 ISS Consensus Workshop participants. Consensus statement on the diagnosis and treatment of children with idiopathic short stature: a summary of the Growth Hormone Research Society, the Lawson Wilkins Pediatric Endocrine Society, and the European Society for Paediatric Endocrinology Workshop. Journal of Clinical Endocrinology and Metabolism 2008. 93 4210–4217. ( 10.1210/jc.2008-0509) Done (Lines 84-89)

Line 115- "Average duration of therapy was 4.5 years (3.5; 9.5)." It is a result, not a method. 

We agree with the reviewer. It has been moved to results

Line 200 - GH secretion when the test was repeated after 1-6 months treatment stop. Done

Line 214 - IGF-BP3. Correct, please. Done

Reviewer 2 Report

I have reviewed the manuscript that presents a retrospective analysis of adult height in patients with isolated GH deficiency (iGHD) who were treated with somatropin (n=24, cases) or with idiopathic short stature (ISS) who received no treatment (n=28, controls). All had IGF-1 < -1.5 SDS. The authors concluded that adult height and stature gain was similar and questioned the efficacy of GH treatment and/or the accuracy of diagnosing iGHD.

The topic is interesting as it raises the discussion about the diagnostic challenge of iGHD. However, the study has some flaws.

Why only patients with IGF-1 <-1,5 SDS were included in the control group?

Why did the authors use the cut-off point < 10 mg/L for diagnosis of GHD since the assay used to dose GH was a sensitive immunometric assay? How many patients in the control group had GH values below this cut-off point?

Patients with GHD generally have lower IGF-1 levels, slower growth velocity, and greater bone age delay than non-GHD patients. However, in the present study, no statistically significant difference in IGF-1 SDS, bone age, bone age delay, and growth velocity was observed between the groups. The authors have to discuss these results.

Although there was no significant difference between the groups by age or sex, case group had a higher proportion of prepubertal children compared to controls. Why? Is it suggestive that many patients in this group was misdiagnosed as GHD and would actually have constitutional growth retardation? The authors have to discuss these results.

GHD is associated with significant growth retardation and delayed bone maturation. On the other hand, puberty is associated with growth spurt and bone age advancement. Why was not observed a significant difference in these parameters between the groups? These are findings that need to be further clarified and commented on by the authors.

Author Response

Responses to comments and queries of Reviewer 2:

I have reviewed the manuscript that presents a retrospective analysis of adult height in patients with isolated GH deficiency (iGHD) who were treated with somatropin (n=24, cases) or with idiopathic short stature (ISS) who received no treatment (n=28, controls). All had IGF-1 < -1.5 SDS. The authors concluded that adult height and stature gain was similar and questioned the efficacy of GH treatment and/or the accuracy of diagnosing iGHD.

The topic is interesting as it raises the discussion about the diagnostic challenge of iGHD. However, the study has some flaws.

Why only patients with IGF-1 <-1,5 SDS were included in the control group?

We have previously observed that the IGF-1 value of -1,5 SDS is the best cut-off to discriminate GHD patients from controls (Ibba et al., reference n. 20). In order to have two matched groups of patients and controls we decided to use this IGF-1 SDS cut-off to select cases and controls with higher probability of being GH deficient. A comment was added in lines 75-81 and 254-259

Why did the authors use the cut-off point < 10 mg/L for diagnosis of GHD since the assay used to dose GH was a sensitive immunometric assay? How many patients in the control group had GH values below this cut-off point?

At the time of testing the cu-off was 10 g/L, therefore we chose that as our cutoff. (Growth Hormone Research Society. Consensus guidelines for the diagnosis and treatment of growth hormone (GH) deficiency in childhood and adolescence: summary statement of the GH Research Society. GH Research Society. J. Clin. Endocrinol. Metab. 85, 3990–3993 (2000).) GH values below the aforementioned cut-off were observed in 4 patients in the control group. All these patients had a second stimulation test with a peak GH ≥ 10 mg/L.A comment was added in lines 53-60

Patients with GHD generally have lower IGF-1 levels, slower growth velocity, and greater bone age delay than non-GHD patients. However, in the present study, no statistically significant difference in IGF-1 SDS, bone age, bone age delay, and growth velocity was observed between the groups. The authors have to discuss these results.

Our findings are consistent with other reports which indicate that children with ISS and IGHD have similar growth responses and do seem to benefit from rhGH therapy. Our results are notably in line with a previous study in which the only difference between cases and controls was the GH peak, which was lower in GHD, while AH and height increases, were similar. We specifically showed that bone age and bone age delay were comparable between the two groups. Once more, this may be due to the low accuracy of stimulation testing and IGF-1 measurement. Since GHD is generally linked to substantial growth retardation and delayed bone maturation, we assumed that GHD may have been misdiagnosed. (see lines 295-299)

Although there was no significant difference between the groups by age or sex, case group had a higher proportion of prepubertal children compared to controls. Why? Is it suggestive that many patients in this group was misdiagnosed as GHD and would actually have constitutional growth retardation? The authors have to discuss these results.

We realized there was an error in the data on pre-pubertal and pubertal patients. There is indeed no difference. We apologize for that! It is extremely challenging to differentiate CGD from IGHD One approach might be, according to the 2019 GH Research Society's recommendation, to use sex steroid priming prior to GHST in peripubertal age. But still there is no consensus on the selection of patients, timing, dose, preparation, and administration of sex steroids. We do not consider priming a routine procedure even in these conditions. As a matter of fact, recent reports indicate that priming is not widely used throughout European centers of pediatric endocrinology. Priming may result in an artificial and transient increase in GH production. (see lines 61-72)

GHD is associated with significant growth retardation and delayed bone maturation. On the other hand, puberty is associated with growth spurt and bone age advancement. Why was not observed a significant difference in these parameters between the groups? These are findings that need to be further clarified and commented on by the authors.

Our results show that children with ISS and IGHD had comparable growth responses and do not seem to benefit from rhGH therapy. We showed that the two groups were comparable in terms of bone age and bone age delay. The low accuracy of stimulation testing and IGF-1 measurement could be the cause. (see lines 167-168 295-299)

Reviewer 3 Report

Please see the attached review.

Author Response

Responses to comments and queries of Reviewer 3:

Thank you for the opportunity to review the manuscript “Evaluation of adult height in patients with non permanent idiopathic GH deficiency” by Agnese Murianni et al. It refers to an important issue of recombinant human growth hormone (rhGH) treatment efficacy in patients with incomplete GHD in childhood. I find the paper interesting, especially for clinicians. However, there are some issues that need to be clarified.

  1. Introduction is too short. I would recommend adding information regarding genetic background of GHD, controversies in GH stimulation tests interpretation (e.g. cut-off 7 ng/mL, sex steroids priming).

We have modified the introduction accordingly (see lines 42-48 53-60)

  1. The groups are small, therefore the results are not explicit. There is no information regarding the number of patients with severe (GH peak < 5 ng/mL) or partial (GH peak 5-10 ng/mL) GHD. If these groups are sufficiently numerous the data should be compared separately between them and controls.

Sixteen patients have partial GHD (peak GH 5–10 g/L) and 8 have severe GHD (peak GH <5 g/L). The patients with partial GHD were evaluated separately from the controls and compared to them. The results obtained are similar to the comparison between 24 cases and controls, as shown in Table 2. (see lines 178-180)

  1. Was the IGF-1 interpreted also in respect of the bone age?

No.

  1. If the measurement of AH in some patients was taken at home please include the exact number of patients.

27 patients were measured AH at home or in a pharmacy

  1. Please use “rhGH” in term of the treatment.

Done

Round 2

Reviewer 2 Report

I have reviewed the manuscript. The subject is relevant and the new version is suitable for publication

Author Response

We hope the manuscript can now be accepted for publication.

Reviewer 3 Report

Thank you for the improvement of the manuscript.

As 2/3 of the GHD group presented with partial GHD it should be marked in the Conclusions.

These patients in general seem to be the most clinically similar to children and adolescents with ISS or constitutional delay of growth and puberty.

Moreover, to make the results and conclusions valuable, the patients with complete GHD should be compared with the controls, as well as with partial GHD group, separately.

Author Response

a) As 2/3 of the GHD group presented with partial GHD it should be marked in the Conclusions.

Done

b) These patients in general seem to be the most clinically similar to children and adolescents with ISS or constitutional delay of growth and puberty.

We agree with the reviewer. We have underlined this concept in lines 224-226 and 308

c) Moreover, to make the results and conclusions valuable, the patients with complete GHD should be compared with the controls, as well as with partial GHD group, separately.

Done
